# Identification of Intercellular Crosstalk between Decidual Cells and Niche Cells in Mice

**DOI:** 10.3390/ijms22147696

**Published:** 2021-07-19

**Authors:** Jia-Peng He, Qing Tian, Qiu-Yang Zhu, Ji-Long Liu

**Affiliations:** Guangdong Laboratory for Lingnan Modern Agriculture, College of Veterinary Medicine, South China Agricultural University, Guangzhou 510642, China; stevencapeng@sina.com (J.-P.H.); tianqing0528@163.com (Q.T.); zqy_961020@163.com (Q.-Y.Z.)

**Keywords:** decidualization, mouse, single-cell RNA-seq, transcriptional changes

## Abstract

Decidualization is a crucial step for human reproduction, which is a prerequisite for embryo implantation, placentation and pregnancy maintenance. Despite rapid advances over recent years, the molecular mechanism underlying decidualization remains poorly understood. Here, we used the mouse as an animal model and generated a single-cell transcriptomic atlas of a mouse uterus during decidualization. By analyzing the undecidualized inter-implantation site of the uterus as a control, we were able to identify global gene expression changes associated with decidualization in each cell type. Additionally, we identified intercellular crosstalk between decidual cells and niche cells, including immune cells, endothelial cells and trophoblast cells. Our data provide a valuable resource for deciphering the molecular mechanism underlying decidualization.

## 1. Introduction

Decidualization is a crucial step for human reproduction. It is a prerequisite for embryo implantation, placentation and pregnancy maintenance [1]. Decidualization initiates spontaneously in the secretory phase of the menstrual cycle, which is tightly controlled by ovarian estrogen and progesterone [2]. Upon decidualization, uterine stromal cells change into large epithelioid cells and secrete two protein markers, decidual prolactin (dPRL) and insulin-like growth factor-binding protein 1 (IGFBP1) [3]. Impaired decidualization may lead to intrauterine growth restriction (IUGR), repeated pregnancy loss (RPL) and severe pre-eclampsia (sPE) [4]. Therefore, it is imperative to understand the molecular mechanism of decidualization.

Due to ethical restrictions, studies on human decidualization are limited to the in vitro level. The in vivo investigation of decidualization heavily relies on mice. Slightly different from humans, the decidual reaction in mice is an embryo-dependent process. Prolactin family 8 subfamily a member 2 (Prl8a2), a paralog of human dPRL, is a well-established marker for mouse decidualization [5]. By using uterus-specific gene knockout mice, a number of genes have been proved to be required for decidualization [6]. Alternatively, several studies have analyzed global gene expression changes associated with decidualization by using high-throughput transcriptomic approaches [5,7,8,9,10]. The limitation of these studies is that the bulk tissue was used. The decidua is a complex tissue consisting of many cell types, including decidual cells, stromal cells, endothelial cells and various immune cells. Thus, bulk-tissue methods were unable to accurately capture cell-type-specific gene expression changes. Moreover, the contribution of crosstalk between different cell types was ignored.

With advances in the single-cell RNA-seq techniques, it is now possible to analyze the global gene expression profile within highly heterogeneous tissues at a single-cell level [11]. In the present study, by using the state-of-the-art single-cell RNA-seq approach, we resolved all the cell types at the implantation site (decidualized) and the inter-implantation site (undecidualized, served as control) of the mouse uterus on gestational day eight. Consequently, we were able to identify differential expression changes associated with decidualization in all the cell types. Additionally, we predicted intercellular crosstalk between the decidual cells and the niche cells. Our study provides a valuable resource for understanding the molecular mechanisms underlying decidualization.

## 2. Results

### 2.1. A Single-Cell Atlas of Mouse Uterus on Gestational Day Eight

To create a cell-type resolved map of a mouse uterus upon decidualization, we performed single-cell RNA-seq analysis (Figure 1A). The implantation site (IS, decidualized uterus) and the inter-implantation site (IIS, undecidualized uterus, served as a control) were collected from gestational day eight (GD8) (Figure 1B). The whole uterus, which consists of the endometrium, the myometrium and the perimetrium, was subjected to single-cell dissociation (Figure 1C). The embryo at the IS was also kept. Single-cell RNA-seq data were generated by using the 10× Genomics platform. After quality control, a total of 15,087 cells (6793 for the IIS and 8294 for the IS) were obtained (Figure 1D,E). In order to validate this single-cell RNA-seq dataset, we also generated a bulk RNA-seq dataset using the same samples (Figure 1F). The cell-averaged single-cell RNA-seq data were highly accordant with the conventional bulk RNA-seq data (r = 0.7465 for the IIS and r = 0.7653 for the IS), indicative of the high quality of our single-cell RNA-seq data.

Unsupervised clustering analysis revealed 21 distinct cell clusters for all the cells from the IIS and the IS combined (Figure 2A). The major cell types were defined using the expression of known cell type-specific genes, with hormone-responsive cells expressing Pgr and Esr1 [12,13], endothelial cells expressing Pecam1 [14] and immune cells expressing Ptprc [15] (Figure 2B).

Hormone-responsive cells included epithelial cells expressing Epcam and Krt19 [16] (Figure 2C), stromal cells expressing Hoxa11 [17] (Figure 2D), smooth muscle cells expressing Acta2 [18] and pericytes expressing Rgs5 [19] (Figure 2E). We found two epithelial cell clusters, LE and GE. LE was luminal epithelial cells expressing Tacstd2 and GE was glandular epithelial cells expressing Foxa2 [20]. We identified five stromal cell clusters, S1, S1p, S2, S3 and S3p. The cells in S2 but not S1 expressed high levels of Hand2, implying that S2 was superficial stromal cells and S1 was deep stromal cells [21]. S1p was a subset of proliferating S1 with a high level of Mki67 (Figure 2G). Notably, there was no proliferating subset for S2. S3 was decidual cells expressing decidualization marker genes Wnt4, Bmp2 and Prl8a2 [22,23,24]. S3p was intermediate decidual cells expressing proliferation marker gene Mki67 (Figure 2G). Only one smooth muscle cell cluster and one pericyte cluster were found.

Endothelial cells had four clusters: VEC and its proliferating subset VECp were vascular endothelial cells expressing Sox17 [14], while LEC and its proliferating subset LECp were lymphatic endothelial cells expressing Prox1 [14] (Figure 2H).

There were eight immune cell clusters (Figure 2I,J). Included were natural killer cells (NK, Ptprc^+^Nkg7^+^Cd3e^−^) [15], proliferating natural killer cells (NKp, Ptprc^+^Nkg7^−^Cd3e^+^Mki67^+^), T cells (T, Ptprc^+^Nkg7^−^Cd3e^+^) [15], B cells (B, Ptprc^+^Cd79a^+^Ms4a1^+^) [15], macrophages (M, Ptprc^+^Adgre1^+^) [25], dendritic cells (DC, Ptprc^+^Itgax^+^) [25], proliferating dendritic cells (DC, Ptprc^+^Itgax^+^Mki67^+^) and plasmacytoid dendritic cells (pDC, Ptprc^+^Siglech^+^) [26].

Finally, we aimed to discover novel markers for each cell type. We selected the genes that were expressed significantly higher in the cell type of interest than the other cell types using the Wilcoxon rank sum test. A complete list of marker genes is presented in Appendix A.

### 2.2. Reconstruction of Decidual Cell Trajectory Across Stromal Cells

In our single-cell RNA-seq data, we identified the following five clusters of stromal cells: S1 (deep stromal cells), S1p (proliferating deep stromal cells), S2 (superficial stromal cells), S3p (intermediate decidual cells) and S3 (decidual cells). We selected signature genes for each cell cluster by using the Wilcoxon rank sum test. After the removal of redundancy, we identified a total of 2113 signature genes (Appendix A). Through a heatmap, we grouped all these signature genes into four gene sets (Figure 3A). GeneSet#1, with 359 genes, was S2-specific. Gene ontology analysis showed that these genes were enriched in developmental processes (*p* = 1.2 × 10^−10^), RNA metabolism (*p* = 4.4 × 10^−9^), protein metabolism (*p* = 1.1 × 10^−5^), cell cycle and proliferation (*p* = 0.00017), cell adhesion (*p* = 0.014) and cell death (*p* = 0.038). GeneSet#2, with 475 genes, was S1-specific. These genes were enriched in cell adhesion (*p* = 1.6 × 10^−11^), developmental processes (*p* = 4.4 × 10^−10^), stress response (*p* = 9.0 × 10^−7^), cell organization and biogenesis (*p* = 0.00019), cell death (*p* = 0.00095) and protein metabolism (*p* = 0.0070). Geneset#3 of 562 genes gradually decreased during the decidualization process. Based on GO, the enriched terms were cell cycle and proliferation (*p* = 2.1 × 10^−11^), protein metabolism (*p* = 5.9 × 10^−9^), cell organization and biogenesis (*p* = 1.2 × 10^−8^), DNA metabolism (*p* = 2.3 × 10^−6^) and transport (*p* = 0.0021). GeneSet#4 of 717 genes gradually increased during the decidualization process. The enriched GO terms were transport (*p* = 8.7 × 10^−9^), protein metabolism (*p* = 0.016) and cell cycle and proliferation (*p* = 0.038). Hsd11b2, C3, Mki67 and Hand2 were representative genes for gene sets 1–4, respectively (Figure 3B).

To further reveal the relationship between these five stromal cell clusters, pseudotime trajectory analysis was conducted by using the Monocle2 software [27]. The cells were arranged in a pseudotime manner with a pedigree reconstruction algorithm for biological processes based on transcriptional similarity. We found that all the stromal cell clusters formed a continuous trajectory with two arms: (1) S1-to-S2 transition, i.e., S1->S2; and (2) decidualization process, i.e., S1->S1p->S3p/S3 (Figure 3C–E).

We are particularly interested in GeneSet#4, the S3-enriched genes. On GD8, the entire IS, from the anti-mesometrial region to the mesometrial region, is fully decidualized. Meanwhile, the IIS shows no sign of decidualization. Thus, the S3-enriched genes could be validated using qTR-PCR with bulk tissues. The top 12 S3-enriched genes (A2m, Dmkn, Mt4, Tdo2, Mt3, Rrm2, Dio3, Serpinb6b, Ass1, Cystm1, Psca and Sfrp5) were selected (Appendix A). Although there was variation in the fold changes between the qRT-qPCR and the bulk RNA-seq, the up-regulation expression trends for all the genes tested were coincident between these two techniques (Appendix A). These data provide the validity of our single-cell RNA-seq data.

### 2.3. Cell-Cell Communication between Decidual Cells and Immune Cells

We investigated the abundance of each immune cell type at the IS compared to the IIS. For each cell type, proliferating and non-proliferating cells were summed. The χ^2^ test was employed to assess the significance of difference between the two groups. By using the criteria of *p* < 0.05 and fold change > 2, the proportions of all the cell types are unchanged, except DC and B cells (Figure 4A).

We investigated the breadth of transcriptional changes in each immune cell type by performing differential gene expression analysis. The proliferating cells were excluded. Using a logFC cutoff of 0.25 and a *p* value cutoff of 0.05, we identified 278, 380, 123, 181, 432 and 270 differentially expressed genes for T, B, NK, M DC and pDC, respectively (Figure 4B and Appendix A). We then explored the biological implications of differentially expressed genes using gene ontology (GO) analysis. A complete list of the enriched GO terms is provided in Figure 4C. These data indicated that each immune cell type invokes distinct biological processes upon decidualization.

We then used the CellChat software [28] to predict the ligand-receptor interactions between the decidual cells and the immune cells. We found a total of 68 ligand-receptor interaction pairs (Figure 5A). Pathway analysis revealed that these ligand-receptor interactions were enriched among the ECM-receptor interaction (FDR = 1.00 × 10^−39^), the PI3K-Akt signaling pathway (FDR = 1.00 × 10^−34^), the Cytokine–cytokine receptor interaction (FDR = 1.00 × 10^−16^), the regulation of actin cytoskeleton (FDR = 1.00 × 10^−14^), the MAPK signaling pathway (FDR = 5.01 × 10^−8^), the Rap1 signaling pathway (FDR = 1.26 × 10^−7^), endocytosis (FDR = 1.58 × 10^−6^), the natural killer cell-mediated cytotoxicity (FDR = 3.98 × 10^−6^), phagosome (FDR = 6.31 × 10^−6^), the TGF-beta signaling pathway (FDR = 1.58 × 10^−5^), the Hippo signaling pathway (FDR = 2.51 × 10^−4^), the NOD-like receptor signaling pathway (FDR = 3.98 × 10^−4^), the Toll-like receptor signaling pathway (FDR = 3.98 × 10^−4^), the mTOR signaling pathway (FDR = 2.00 × 10^−3^) and the Ras signaling pathway (FDR = 2.00 × 10^−4^) (Figure 5B).

### 2.4. Cell-Cell Communication between Decidual Cells and Endothelial Cells

By using the criteria of *p* < 0.05 and fold change > 2, we found that the proportion of LEC was unchanged, whereas the proportions of VEC were significantly increased (Figure 6A). Using a logFC cutoff of 0.25 and a *p* value cutoff of 0.05, we identified 627 and 111 differentially expressed genes for VEC and LEC, respectively (Figure 6B and Appendix A). The differentially expressed genes were further characterized using GO analysis (Figure 6C).

We predicted the ligand-receptor interactions between decidual cells and VEC/LEC. We found a total of 101 ligand-receptor interaction pairs for decidual-VEC crosstalk and 104 ligand-receptor interaction pairs for decidual-LEC crosstalk, respectively (Figure 7A). Pathway analysis revealed that these ligand-receptor interactions were enriched among the PI3K-Akt signaling pathway (FDR = 1.00 × 10^−56^), the ECM-receptor interaction (FDR = 1.00 × 10^−46^), the Rap1 signaling pathway (FDR = 1.00 × 10^−20^), the Cytokine-cytokine receptor interaction (FDR = 1.00 × 10^−20^), the Ras signaling pathway (FDR = 1.00 × 10^−17^), the regulation of actin cytoskeleton (FDR = 1.00 × 10^−15^), the HIF-1 signaling pathway (FDR = 1.26 × 10^−9^), the MAPK signaling pathway (FDR = 1.58 × 10^−9^), endocytosis (FDR = 6.31 × 10^−8^), the Phospholipase D signaling pathway (FDR = 3.16 × 10^−6^), the Hippo signaling pathway (FDR = 3.98 × 10^−6^), the TGF-beta signaling pathway (FDR = 2.00 × 10^−5^), the mTOR signaling pathway (FDR = 5.01 × 10^−4^), phagosome (FDR = 1.26 × 10^−3^), the NOD-like receptor signaling pathway (FDR = 6.31 × 10^−3^) and the Toll-like receptor signaling pathway (FDR = 6.31 × 10^−3^) (Figure 7B).

### 2.5. Cell-Cell Communication between Decidual Cells and Trophoblast Cells

Trophoblast cells may interact with decidual cells and pay an important role during the decidualization process. However, due to the relatively small number of embryonic cells at the implantation site, we did not find any fetal cell clusters in our single-cell RNA-seq data. Alternatively, we re-analyzed a published single-cell RNA-seq dataset on mouse E7.5 embryos (GSM3457437) [29]. E7.5 is equivalent to gestational day eight in our study. By using marker gene Plac1 [30], we discovered a trophoblast cell cluster (Figure 8A). These cells were further divided into ectoplacental cone (EPC) expressing Dlx3 [31] and trophoblast giant cells (TGC) expressing Prl3d1 [32] (Figure 8B).

By using the CellChat software [28], we generated an interaction network underlying decidual cells and EPC/TGC (Figure 8C). We found a total of 38 ligand-receptor interaction pairs for decidual-EPC crosstalk and 36 ligand-receptor interaction pairs for decidual-TGC crosstalk (Figure 8D). Based on pathway analysis, these ligand-receptor interactions were enriched among the ECM-receptor interaction (FDR = 1.00 × 10^−37^), the PI3K-Akt signaling pathway (FDR = 1.00 × 10^−29^), the Cytokine-cytokine receptor interaction (FDR = 1.00 × 10^−11^), the regulation of actin cytoskeleton (FDR = 1.00 × 10^−9^), the Rap1 signaling pathway (FDR = 2.00 × 10^−8^), the TGF-beta signaling pathway (FDR = 3.98 × 10^−6^), the Ras signaling pathway (FDR = 7.94 × 10^−7^), the Phospholipase D signaling pathway (FDR = 7.94 × 10^−5^), the Hippo signaling pathway (FDR = 1.26 × 10^−3^) and the MAPK signaling pathway (FDR = 1.26 × 10^−3^) (Figure 8E).

## 3. Discussion

We have recently established a single-cell atlas of mouse uterus during embryo implantation using the 10× Genomics approach [33]. In this study, we systematically characterized the mouse uterus on GD8 at a single-cell resolution. We identified 21 distinct cell clusters in mouse uterus during decidualization, including 5 stromal cell clusters, 2 epithelial cell clusters, 1 smooth muscle cell cluster, 1 pericyte cluster, 4 endothelial cell clusters and 8 immune cell clusters. To our knowledge, our study is the first comprehensive single-cell transcriptomic atlas for mouse decidualization.

We used 2 mg/mL Collagenase II and 10 mg/mL Dispase II for a single-cell suspension preparation. Collagenase II is a crude collagenase preparation with weak trypsin-like activity and, thus, trypsin was not used in our method. It turned out that the cell viability was >80% and the percentage of cell clumps was <10%, indicative of the high efficiency of our method. Nevertheless, we noticed that the percentage of decidualized stromal cells was much lower than expected: S3 (decidual cells) was 4.4% and S3p (intermediate proliferating decidual cells) was 1.7%. As we observed lower cell viability at the implantation site compared to the inter-implantation site, we suspected that the collagenase treatment might selectively destroy the decidualized cells. This was in line with the finding that the highly expressed decidual-specific marker gene Prl8a2 contaminated all the other cells, indicative of the rupture of the decidual cells during the single-cell suspension preparation. 

Moreover, we would like to note that the collagenase treatment might cause transcriptional disturbances, leading to artifacts [34]. We compared single-cell RNA-seq data against parallel bulk RNA-seq data. We found that a large number of genes were changed in both the IS and the IIS (Appendix A). The complete list of differentially expressed genes is listed in Appendix A. There were 1590 down-regulated genes and 1297 up-regulated genes shared by the IS and the IIS (Appendix A). We focused on up-regulated genes because we suspected that these genes might exhibit de novo expression upon collagenase treatment. We examined the expression pattern of 12 selected up-regulated genes in single-cell RNA-seq data. We found that Egr1, Egr2, Egr3, Fos, Jun and Atf3 were ubiquitously expressed in nearly all cell types, whereas Ccl6, Ccl9, Ccl5, Ccl8, Ar and Tagln were expressed in a cell-type-restricted manner (Appendix A). We believe that these genes could be used as marker genes without any problem, although they are similar to “artificial markers”. However, it will be problematic if they are chosen as differentially expressed genes, because the de novo expression levels might vary across each collagenase treatment. Our data suggest that single-cell RNA-seq data should be interpreted with caution.

In this study, the embryo at the IS was kept. We estimated that there were around 2000 cells per embryo. We usually obtained two million cells per sample in the single-cell dissociation procedure. Approximately 5000 cells were sequenced per sample by the 10× platform. By calculating the probability, we found that only ~5 cell per embryo could be captured in our single-cell RNA-seq data. This was in line with our findings that there were no embryo-derived cell clusters in our single-cell RNA-seq data. The embryo-derived cells, if they existed at all, might be scattered among uterine cells in our single-cell RNA-seq data. Due to their small number, these cells might not have an obvious effect on the assigning cell type label for each cell cluster. Moreover, these cells might not interfere with the process for identifying differentially expressed genes either, because we set min.pct to 0.20, which means that we required the differentially expressed genes to be expressed in a minimum of 20% of cells within the cluster of interest.

In mice, embryo implantation initiates at midnight on GD4. Upon embryo implantation, the superficial stromal cells proximally surrounding the implantation chamber exhibit proliferation and spreading. These cells differentiate and form the primary decidual zone (PDZ) on the afternoon of gestational day five. The PDZ subsequently undergoes apoptosis and disappears by GD8. Meanwhile, the stromal cells next to the PDZ continue to proliferate and differentiate into decidual cells, forming the secondary decidual zone (SDZ). The SDZ is full developed on GD8. Here, we found five stromal cell clusters for the IS (S2, S1, S1p, S3 and S3p) and three stromal cell clusters for the IIS (S2, S1 and S1p). S2 was superficial stromal cells and S1/S1p was deep stromal cells, which is in line with our previous study [33]. S3 and S3p were IS-specific: S3 was decidual cells expressing the decidualization marker genes Wnt4, Bmp2 and Prl8a2, while S3p was intermediate decidual cells expressing the proliferation marker gene Mki67. Through pseudotime trajectory analysis, we found that S3p/S3 decidual cells originated from S1/S1p, but not from S2 cells. By using the Wilcoxon rank sum test, we identified two gene sets (GeneSet#3 and GeneSet#4) associated with decidualization. GeneSet#3 was gradually decreased during the decidualization process, whereas GeneSet#4 was gradually increased during the decidualization process. Based on GO analysis, the enriched terms for GeneSet#3 were cell cycle and proliferation, protein metabolism, cell organization and biogenesis, DNA metabolism and transport. The enriched GO terms for GeneSet#4 were transport, protein metabolism and cell cycle and proliferation. Our data provided clues for the development trajectory for decidualized stromal cells in SDZ.

We are particularly interested in GeneSet#4, which may represent the key molecular mechanism of decidualization. The best strategy to validate our findings is to obtain a pure cell population by using fluorescence activated cell sorting (FACS). Previously, CD13-specific antibodies were used to sort stromal cells by FACS from human endometrial biopsies [35,36]. However, the same method to sort decidual cells is yet to be established. GeneSet#4 was S3-enriched genes. On GD8, the IS but not the IIS is fully decidualized. Thus, the genes in GeneSet#4 should be up-regulated in the IS compared to the IIS at the bulk-tissue level. In this way, we were able to validate GeneSet#4, at least in part, by using qRT-PCR. The top 12 genes were tested. It turned out that all these genes were up-regulated in the IS compared to the IIS. This result might provide the validity of our single-cell RNA-seq data.

Immune cells play an important role in embryo implantation and decidualization in mice [37]. We investigated the abundance of each immune cell type at the IS compared to the IIS. By using the criteria of *p* < 0.05 and fold change > 2, the proportions of all the cell types are unchanged, except for DC and B cells. As B cells were a minor cell type (< 5%) in the uterus, we thus focused on DC cells. It has been demonstrated that the depletion of DC cells resulted in a severe impairment of the decidualization process, leading to embryo resorption [38]. We found that S3 decidual cells expressed secreted factors such as Tgfb2, Tgfa, Ptn, Mif, Mdk, Lgals9, Igf1, Fgf2 and C3, while their receptors were expressed on DC cells. Thus, these secreted factors might attract DC cells, leading to the accumulation of DC cells in the decidualized implantation site of the uterus. On the other hand, DC cells expressed secreted factors Tnf, Tgfb1, Nampt and Mif. As their receptors were expressed on S3 decidual cells, these secreted factors might contribute to the decidualization reaction of stromal cells.

Endothelial cells are in the niche of decidual cells. We hypothesized that endothelial cells might contribute to decidualization. We found that VEC cells were significantly increased, while LEC cells were slightly decreased upon decidualization. This was in line with the fact that the decidualization process is accompanied by angiogenesis [39]. Though cell–cell communication analysis, we found that angiogenesis was likely mediated by Vegfa and Vegfb expressed on S3 decidual cells and their receptors Vegfr1 and Vegfr2 expressed on VEC cells. VEC cells might promote decidualization by expressing soluble factors such as Hbegf. The contribution of VEC to decidualization deserves further investigation.

Previously, E7.5 embryos obtained from the GD8 uterus have been subjected to single-cell RNA-seq [29]. Through data integration, we inferred cell–cell communication between E7.5 embryos and the implantation site of the GD8 uterus by their expression of ligand-receptor pairs. We generated an interaction network for decidual cells from the uterus and EPC/TGC cells from the embryo. These interactions might provide clues for the physical function of decidual cells in placentation and fetal development.

In summary, we revealed intercellular crosstalk between decidual cells and niche cells, including immune cells, endothelial cells and trophoblast cells. This cell atlas of mouse uterus on GD8 provides an essential resource for understanding the molecular mechanism underlying decidualization.

## 4. Materials and Methods

### 4.1. Sample Collection

Adult CD-1 mice of the SPF grade were used in this study. All mice were caged under light-controlled conditions (14 h/10 h light/dark cycles) with free access to regular food and water. Female mice were mated with fertile males and mating was confirmed the next morning by the presence of a vaginal plug. The day of the vaginal plug was denoted as gestation day 1 (GD1). On GD8, the implantation sites (decidualized uterus) and non-implantation sites (undecidualized uterus, served as a control) were collected separately.

### 4.2. Bulk RNA-Seq Analysis

The total RNA from uterine tissues were extracted with the TRIzol reagent (Invitrogen). RNA-seq libraries were generated by using the TruSeq RNA sample preparation kit (Illumina, San Diego, CA, USA) and sequenced on an Illumina HiSeq 2500 system with the paired-end 150-bp protocol. Raw data were analyzed using TopHat v2.0.4 [40] and Cufflinks v2.2.1 [41]. The mouse genome UCSC mm10 was used for reference. Gene expression levels were measured as fragments per kilobase per million (FPKM).

### 4.3. Single-Cell Dissociation of Mouse Uterus

Uterine samples from 3 mice for each group were pooled and minced with a blade. Samples were then incubated in dissociation buffer containing 2 mg/mL Collagenase II (#C6885, Sigma-Aldrich, St. Louis, MO, USA), 10 mg/mL Dispase II (#354235, Corning, Corning, NY, USA) and 50,000 U/mL DNase I (#DN25, Sigma-Aldrich, St. Louis, MO, USA) for up to 30 min at 37 °C in a shaking incubator. The digestion progress was checked every 5 min with a microscope until a single cell suspension was achieved. The single-cell suspension was then passed through a 40-micrometer cell strainer to remove undigested tissues. Cells were spun down at 250× *g* at 4 °C for 4 min and the pelleted cells were washed using centrifugation. In order to measure cell viability, cells were strained with AO/PI solution (#CS2-0106, Nexcelom Bioscience, Lawrence, MA, USA) and counted using a Cellometer Auto 2000 instrument (#SD-100, Nexcelom Bioscience, Lawrence, MA, USA). The single-cell suspension was carried forward to single-cell RNA-seq only if the cell viability was >80% and the percentage of cell clumps was <10%.

### 4.4. Single-Cell RNA-Seq Library Preparation and Sequencing

The final concentration of single-cell suspension was adjusted to 1000 cells/μL and a volume of 15 µL was loaded into one channel of the Chromium^TM^ Single Cell B Chip (#1000073, 10× Genomics, Pleasanton, CA, USA), aiming at recovering 8000–10,000 cells. The Chromium Single Cell 3′ Library and Gel Bead Kit v3 (#1000075, 10× Genomics, Pleasanton, CA, USA) was used for single-cell bar-coding, cDNA synthesis and library preparation, following the manufacturer’s instructions. Library sequencing was performed on an Illumina NovaSeq 6000 system configured with the paired-end 150-bp protocol at a sequencing depth of approximately 400 million reads.

### 4.5. Single Cell RNA-Seq Data Analysis

Raw data (bcl files) from the Illumina NovaSeq 6000 platform were converted to fastq files using the bcl2fastq tool (v2.19.0.316, Illumina, San Diego, CA, USA). These fastq files were aligned to the UCSC mm10 mouse reference genome by using the CellRanger software (v3.0.1, 10× Genomics, Pleasanton, CA, USA). The resulting gene counts matrix was analyzed with the R package Seurat (v3.1.3) [42]. Cell with fewer than 200 or greater than 6000 unique genes, as well as cells with greater than 25% of mitochondrial counts, were excluded. Meanwhile, genes expressed in fewer than 3 cells were removed. Following data filtering, the gene counts matrix was normalized and scaled by using the NormalizeData and ScaleData functions, respectively. The top 2000 highest variable genes were used for the principal component analysis (PCA) and the optimal number of PCA components was determined using the JackStraw procedure. Single cells were clustered using the K-nearest neighbor (KNN) graph algorithm in PCA space and visualized using the t-distributed stochastic neighbor embedding (tSNE) dimensional reduction technique. The cell type label for each cell cluster was manually assigned based on canonical cell markers. The FindAllMarkers function was used to identify novel marker genes for each cluster with min.logfc being set to 0.25 and min.pct being set to 0.20. By using the same parameters, the FindMarkers function was used to find differential expressed genes in the IS compared to the IIS for each cell type.

### 4.6. Pseudotime Analysis

The Monocle2 package v2.18.0 was used for pseudotime analysis [27]. The count data and meta data were exported from the Seurat object and then imported into the CellDataSet object in Monocle2. Feature genes were selected by using the differentialGeneTest function. After dimension reduction by using the DDRTree algorithm, the orderCells function was used to infer the trajectory with default parameters. The reconstructed trajectory was visualized using the plot_cell_trajectory function.

### 4.7. Gene Ontology Analysis

Gene ontology (GO) analysis was performed as described previously [43]. GO terms were grouped according to the biological process category defined in the Mouse Genome Informatics (MGI) GOslim database [44]. To test for enrichment, a hypergeometric test was conducted and *p* < 0.05 was used as the significance threshold to identify enriched GO terms.

### 4.8. Pathway Analysis

Pathway enrichment analysis was conducted by using the Metascape v7.4 online tools [45]. The significance threshold for FDR was set at 0.05.

### 4.9. Cell-Cell Communication Analysis

The CellChat v1.1.0 R package [28] was used to infer cell–cell communication based on ligand-receptor interaction with default parameters. For each ligand-receptor pair, CellChat assigned a communication probability value by the law of mass action based on the average expression values of a ligand by one cell group and that of a receptor by another cell group. The statistical significance of communication probability values was assessed using a permutation test. *p* < 0.05 was considered significant.

### 4.10. Validation by Quantitative RT-PCR

The TRIzol reagent (Invitrogen, Carlsbad, CA, USA) was used to extract total RNAs. The cDNAs were synthesized with the PrimeScript reverse transcriptase reagent kit (TaKaRa, Dalian, China). A quantitative RT-PCR was performed on Applied Biosystems 7500 (Life Technologies, Carlsbad, CA, USA) with the THUNDERBIRD SYBR qPCR Mix (Toyobo, Osaka, Japan). The Rpl7 gene was used as the reference gene for data normalization. A complete list of primer sequences is provided in Appendix A.

## Figures and Tables

**Figure 1 ijms-22-07696-f001:**
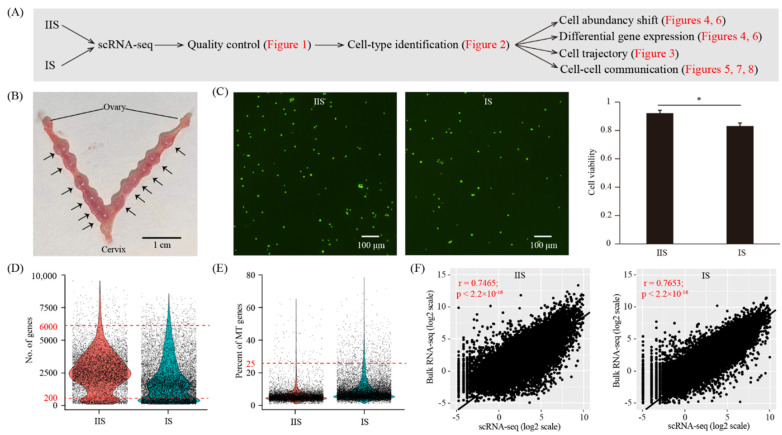
Single-cell transcriptome analysis of the decidualized mouse uterus on gestational day 8. (**A**) A flowchart overview of this study. IIS, inter-implantation site (undecidualized uterus, served as control); IS, implantation site (decidualized uterus). (**B**) A photograph of mouse uterus on gestational day 8. The positions of embryo implantation sites were marked with an arrow. (**C**) Cell viability analysis of single-cell suspension. Cells were stained with the AO/PI solution. Cells in green were live and cells in yellow were dead. A representative photo was provided, and a bar plot was shown on the right. *n* = 3; * *p* < 0.05. (**D**,**E**) Single-cell RNA-seq data pre-processing and quality control. Cells with detected genes of fewer than 200 or more than 6000 were removed. Only cells with total mitochondrial gene expression below 25% were kept. (**F**) Scatter plots showing the correlation between single-cell RNA-seq and bulk RNA-seq. For single-cell RNA-seq data, gene expression levels were averaged and normalized as transcript per million (TPM). For bulk RNA-seq data, gene expression levels were measured as fragments per kilobase per million (FPKM).

**Figure 2 ijms-22-07696-f002:**
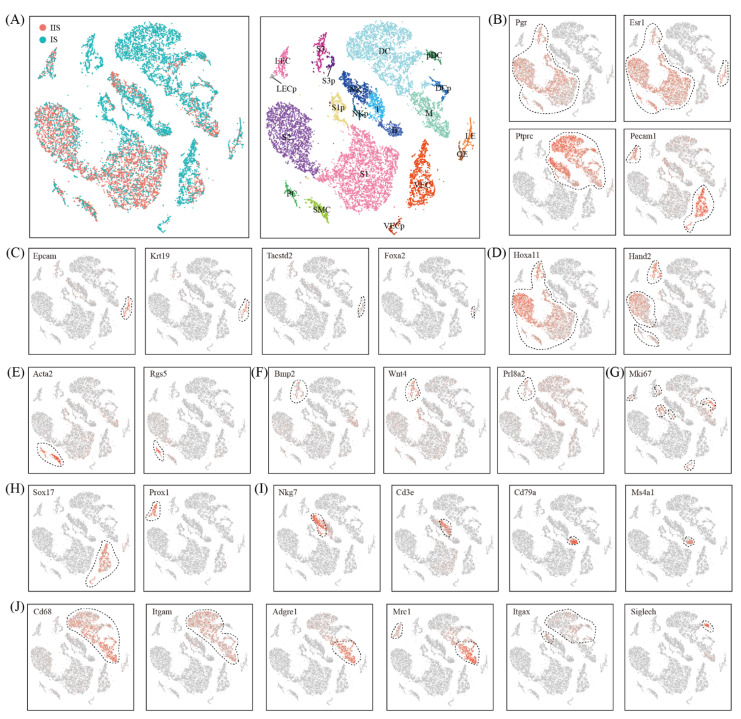
Identification of different cell types in mouse uterus by using canonical marker genes. (**A**) The t-Stochastic neighbor embedding (TSNE) representation of single-cell RNA-seq data obtained from IS and IIS of the uterus. Single cells were grouped by cellular origin (right) and cell clusters (left). LE, luminal epithelial cells; GE, glandular epithelial cells; S1, deep stromal cells; S1p, proliferating deep stromal cells; S2, superficial stromal cells; S3, decidual cells; S3p, proliferating decidual cells; SMC, smooth muscle cells; PC, pericytes; VEC, vascular endothelial cells; VECp, proliferating vascular endothelial cells; LEC, lymphatic endothelial cells; LECp, proliferating lymphatic endothelial cells; NK, natural killer cells; T, T cells; NKp, proliferating natural killer cells; B, B cells; M, macrophages; DC, dendritic cells; DCp, proliferating dendritic cells; pDC, plasmacytoid dendritic cells. (**B**–**J**) The expression pattern of canonical marker genes projected onto TSNE plots. Shown are pan-marker genes for hormone-responsive cells, immune cells and endothelial cells (**B**), epithelial cells (**C**), stromal cells (**D**), smooth muscle cells and pericytes (**E**), decidual cells (**F**), proliferating cells (**G**), endothelial cells (**H**), lymphocytes (**I**) and myeloid cells (**J**). Dashed lines denote the boundaries of the cell cluster of interest.

**Figure 3 ijms-22-07696-f003:**
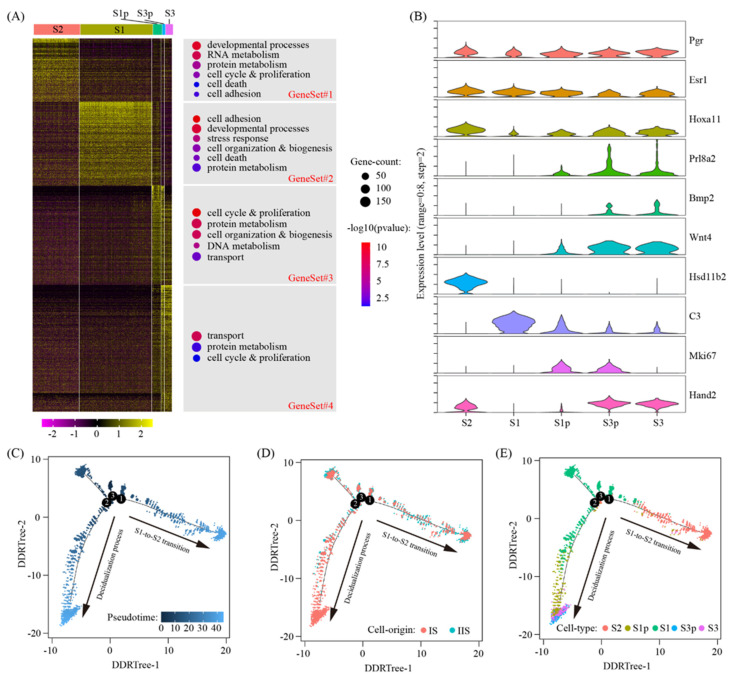
Sub-cluster analysis of stromal cells. (**A**) Heatmap of gene expression signatures for all stromal cell sub-clusters (left). All signature genes were divided into 4 gene sets and enriched gene ontology terms were assigned accordingly (right). (**B**) The expression pattern of marker genes for sub-clusters of stromal cells using violin plot. Shown were pan-stromal cell markers (Pgr, Esr1 and Hoxa11), decidual cell markers (Prl8a2, Wnt4 and Bmp2) and gene set reprehensive genes (Hsd11b2 for #1, C3 for #2, Mki67 for #3 and Hand2 for #4). (**C**–**E**) Pseudotime ordering of stromal cells. The distribution of pseudotime (**C**), cell origin (**D**) and cell type (**E**) across the reconstructed trajectory were displayed.

**Figure 4 ijms-22-07696-f004:**
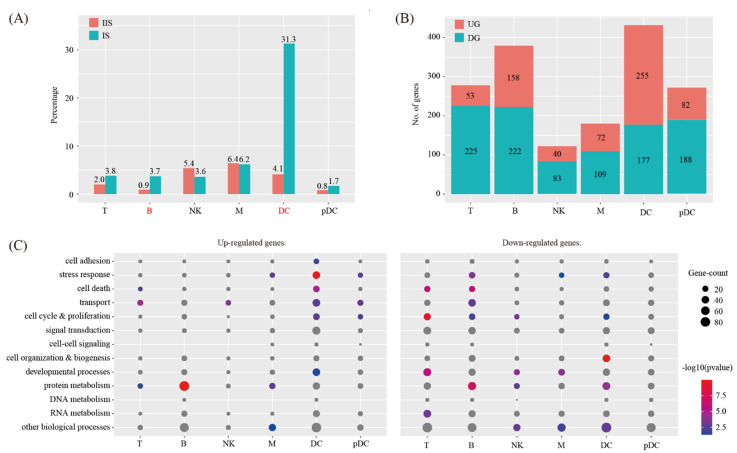
Identification of differentially expressed genes in immune cells upon decidualization. (**A**) Bar plot showing the cell population change of 6 major immune cell types at IS compared to IIS. Cell types with FC > 2 and *p* < 0.05 using χ^2^ test were labeled in red. (**B**) Bar plot showing the count of differentially expressed genes in each immune cell type. The threshold values for differentially expressed genes were logFC > 0.25 and *p* < 0.05. UG, up-regulated gene; DG, down-regulated genes. (**C**) Gene ontology (GO) enrichment analysis of differentially expressed genes. Differentially expressed genes were grouped based on MGI GOslim terms under the biological process categories. Up-regulated genes and down-regulated genes were tested separately. Abbreviations for cell types are listed in Figure 2.

**Figure 5 ijms-22-07696-f005:**
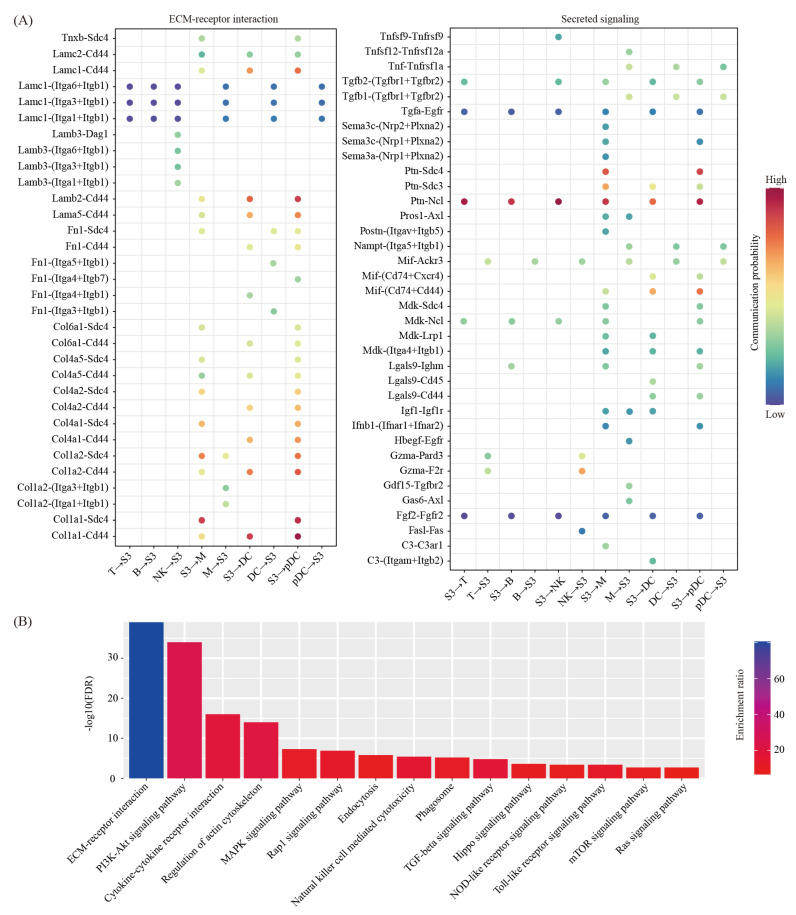
Cell–cell communication between decidual cells and immune cells. (**A**) Dot plot showing selected ligand-receptor interactions underlying crosstalk between decidual cells (S3) and immune cells (T, NK, B, M, DC and pDC). The communication probability defined by the CellChat software were indicated by color. (**B**) KEGG pathway enrichment analysis of ligand-receptor pairs by using Metascape online tools.

**Figure 6 ijms-22-07696-f006:**
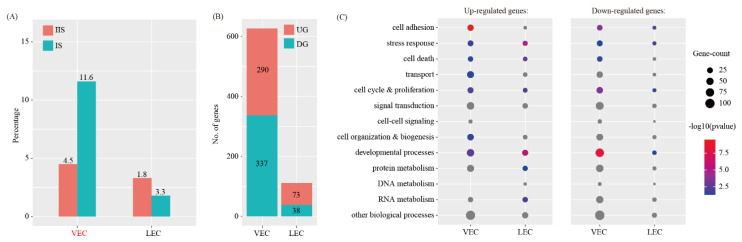
Identification of differentially expressed genes in endothelial cell during decidualization. (**A**) Bar plot showing the cell population change of 2 major endothelial cell types (VEC and LEC) at IS compared to IIS. Cell types with FC > 2 and *p* < 0.05 using χ^2^ test were labeled in red. (**B**) Bar plot showing the count of differentially expressed genes in each immune cell type. The threshold values for differentially expressed genes were logFC > 0.25 and *p* < 0.05. UG, up-regulated gene; DG, down-regulated genes. (**C**) Gene ontology (GO) enrichment analysis of differentially expressed genes. Differentially expressed genes were grouped based on MGI GOslim terms under the biological process categories. Up-regulated genes and down-regulated genes were tested separately.

**Figure 7 ijms-22-07696-f007:**
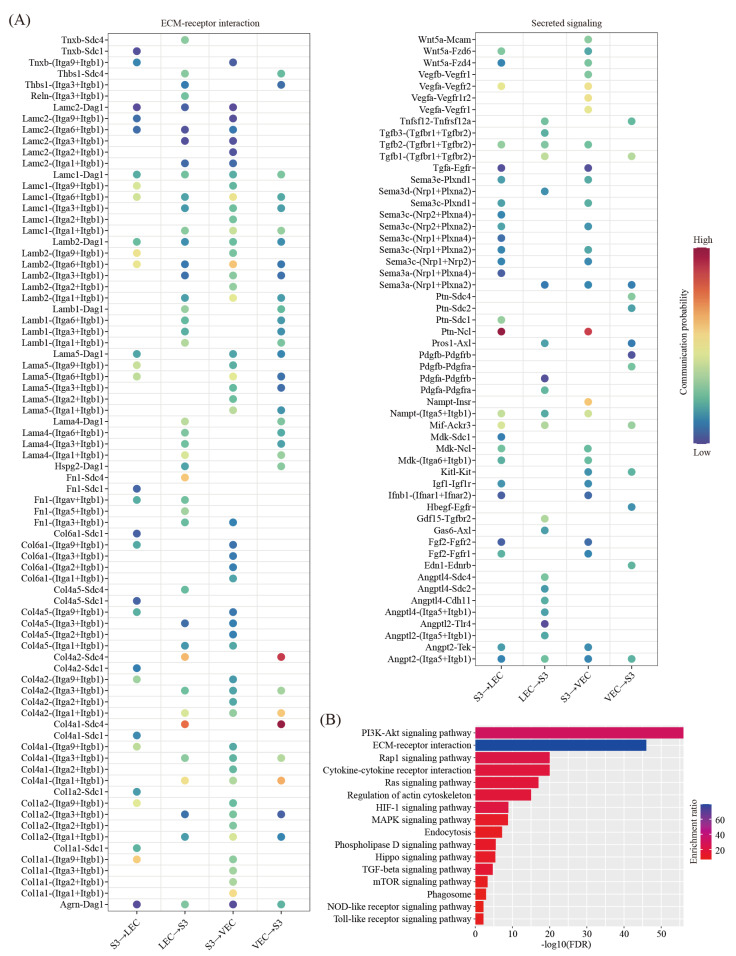
Cell–cell communication between decidual cells and endothelial cells. (**A**) Dot plot showing selected ligand-receptor interactions underlying the crosstalk between decidual cells (S3) and endothelial cells (VEC and LEC). The communication probability defined by the CellChat software were indicated by color. (**B**) KEGG pathway enrichment analysis of ligand-receptor pairs by using the Metascape online tools.

**Figure 8 ijms-22-07696-f008:**
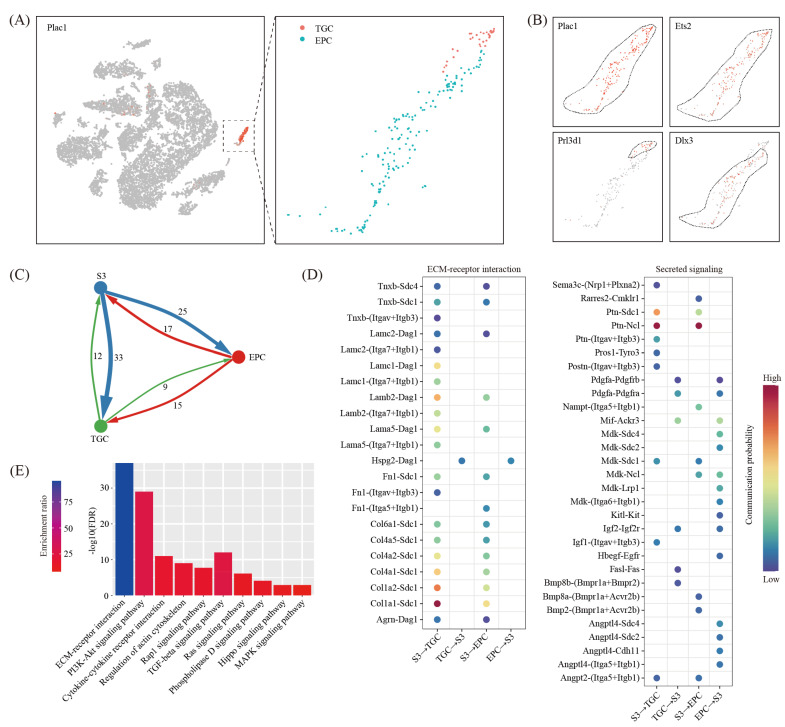
Cell–cell communication between decidual cells and trophoblast cells. (**A**) A single-cell atlas of E7.5 embryos, which were collected from GD8 uterus from a previous study. TSNE clustering was employed (**left**) and Plac1-positive cell were further divided into EPC (ectoplacental cone) and TGC (trophoblast giant cells) (**right**). (**B**) TSNE map showing the expression pattern of well-known marker genes. (**C**) Network plot showing the ligand-receptor interaction events between decidual cells (S3) and blastocyst cells (EPC and TGC). Cell–cell communication was indicated by the connected line. The thickness of the lines is correlated with the total number of ligand-receptor interaction events. (**D**) Dot plot showing selected ligand-receptor interactions underlying the crosstalk between decidual cells and trophoblast cells. The communication probability defined by the CellChat software were indicated by color. (**E**) KEGG pathway enrichment analysis of ligand-receptor pairs by using the Metascape online tools.

## Data Availability

All raw and analyzed sequencing data can be found at Gene Expression Omnibus.

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
