# Peer review of "Identification of Intercellular Crosstalk between Decidual Cells and Niche Cells in Mice"

_ijms, 2021, doi:10.3390/ijms22147696_

Round 1
Reviewer 1 Report
The manuscript by Peng He et al. is focused on single cell analysis of decidualization. It provides extensive atlas of cell types and specifically expressed marker genes. Studied theme is important and results are valuable for specific scientific community. However, there are several important questions to be answered.
- My main comment is about experimental design. Conditions used for tissue dissociation are very risky and high temperature together with absence of transcription inhibitors will lead to massive de novo gene expression of early response genes (fos, jun, egr1 etc.). Authors commented briefly this in the discussion and suggested further investigation. In my opinion, the effect of this technical problem should be presented already in this manuscript, since it can greatly affect presented results. The impact of de novo expression should be presented to trust expression of specific cell marker genes.
- Data sharing. I did not find link to obtained sequences, which should be completely available for readers (after publication). Also most of the results are presented in the form of figures, but there is huge amount of detailed results (such as complete cell type marker lists, comparisons between conditions, complete lists of GO terms), which should be available for readers in the form of supplement tables. This is important especially for manuscript presenting just data from scRNA-Seq.
- I miss data validation. Did authors validated at least several top genes using alternative method? It will be very important since the authors suggested potential loss of certain cell types.
- In several figures (4, 6) GO analysis is presented as colored circles. However some of the points are grey, which is not on the color scale. What does it mean?
- I found several typos especially in the Method section.
Reviewer 2 Report
This is generally interesting article describing global gene expression changes in mouse uterine cells during decidualization process. However there are questions that must be addressed to the authors.
- Genes marker as the basis for cell identification is an acceptable method, although I believe it is not the optimal method. The authors seem to be aware of this fact as evidenced by the comments in the discussion indicating that collagenase treatment might cause transcriptomic disturbances, leading to artifacts (line 273 – 274). Moreover, marker gene expression and protein marker expression on cell surface is sometimes not the same. Therefore, in my opinion fluorescence activated cell sorting (FACS) is the best method to obtain pure cell population.
- There is lack of RNA-seq data validation. Performing either qPCR or protein level detection would be beneficial.
- In the first and last paragraph of the discussion there are fragments (lines 260-261, 329-330, 332-333) that are very similar to those found in the discussion of another article by the authors (Int. J. Mol. Sci. 2021, 22, 5177). Only the word “implantation” was replaced by the word “decidualization”.
Reviewer 3 Report
Thank you for the opportunity to review an interesting manuscript.
This manuscript is very interesting and provides valuable information about the detailed expression profile of decidualized cells in the uterus. Especially they compared various cell types to understand the crosstalk between niche cells. This data is very unique, therefore this manuscript is publishable with a minor change.
In the result section, author mentioned that the embryo at IS was also kept during single-cell dissociation. Is there any effect of embryo on your result? Some different gene expression profiles between IS and IIS might be a consequence of embryonic gene expression. Please explain in the discussion section about any possible effect of embryo in data, or how the author excluded the effect of remaining embryo in IS sample.
Round 2
Reviewer 1 Report
My comments were answered. I recommend accepting the manuscript for publication.
Author Response
Reply: Thank you very much!
Reviewer 2 Report
The manuscript has been improved compared to its first version. However, there are still mistakes in the text. The first and last paragraphs of the discussion in the author’s’ response file have been modified. In contrast, both paragraphs in the manuscript remain unchanged. Reference 33 in lines 263 and 321 and the same reference 33 in line 336 refer to different publications - in the first case, for publication in Int J Mol Sci 2021, 22 (removed from the list), in the second case, for publication in Reprod Biol Endocrinol 2017, 15, 22 (added to the list).
Author Response
Reply: The first and last paragraphs are finally changed this time.
For Reference 33, lines 263 and 321 should be Int J Mol Sci 2021, 22, while line 336 should be Reprod Biol Endocrinol 2017, 15, 22. In this revision, the EndNote-formatted references have been checked manually and this error has been corrected.
Thanks again for your review of our revised manuscript!